# Hypervirulent *Klebsiella pneumoniae* Endogenous Endophthalmitis—A Global Emerging Disease

**DOI:** 10.3390/life11070676

**Published:** 2021-07-10

**Authors:** Dragos Serban, Alina Popa Cherecheanu, Ana Maria Dascalu, Bogdan Socea, Geta Vancea, Daniela Stana, Gabriel Catalin Smarandache, Alexandru Dan Sabau, Daniel Ovidiu Costea

**Affiliations:** 1Faculty of Medicine, “Carol Davila” University of Medicine and Pharmacy Bucharest, 020011 Bucharest, Romania; dragos.serban@umfcd.ro (D.S.); alina.cherecheanu@umfcd.ro (A.P.C.); geta.vancea@umfcd.ro (G.V.); gabriel.smarandache@umfcd.ro (G.C.S.); 24th Department of General Surgery, University Emergency Hospital Bucharest, 050098 Bucharest, Romania; 3Department of Ophthalmology, University Emergency Hospital Bucharest, 050098 Bucharest, Romania; cercetareoftalmo@suub.ro; 4Department of Surgery, “Sf Pantelimon” Emergency Hospital Bucharest, 021659 Bucharest, Romania; 5“Victor Babes” Infectious and Tropical Disease Hospital Bucharest, 030303 Bucharest, Romania; 63rd Clinical Department, Faculty of Medicine, “Lucian Blaga” University Sibiu, 550024 Sibiu, Romania; alexandru.sabau@ulbs.ro (A.D.S.); daniel.costea@365.univ-ovidius.ro (D.O.C.); 7Faculty of Medicine, Ovidius University Constanta, 900527 Constanta, Romania

**Keywords:** hypervirulent *Klebsiella pneumoniae*, invasive liver abscess syndrome, percutaneous drainage, endophthalmitis, early diagnosis, intravitreal antibiotherapy

## Abstract

The review aims to document the new emerging hypervirulent *Klebsiella pneumoniae* (Kp) endogenous endophthalmitis (EKE) in terms of incidence, microbiological characterization of the pathogenic agent, associated risk factors, management, and outcomes. Hypervirulent (hv) strains of KP (hvKp) induce invasive liver abscesses (LA) with specific clinical features. Up to 80–90% of cases have hepatic liver abscess as a primary focus of infection, followed by renal or lung hvKp infections. However, the incidence of EKE in patients with KPLA varied between 3.4% (19) and 12.6% (13), with a total of 95 cases of endophthalmitis in 1455 cases of KPLA (6.5%). Severe visual loss was encountered in 75% of cases, with 25% bilateral involvement. Intravitreal antibiotics are the mainstay therapeutic approach. Pars plana vitrectomy is a subject of controversy. HvKp strains present mostly natural “wild-type” antibiotic resistance profile suggestive for community-acquired infections, being highly susceptive to the third and fourth generation of cephalosporins and carbapenems. Antimicrobial resistance in hypervirulent strains was recently documented via plasmid transfer and may result in extremely difficult to treat cases. Global dissemination of these strains is a major epidemiologic shift that should be considered in the diagnostic and therapeutic management of patients with endogenous endophthalmitis. Ophthalmologic screening in patients with KPLA and other hvKp infections and a multidisciplinary therapeutic approach is extremely important for early diagnosis and preservation of the visual function.

## 1. Introduction

*Klebsiella pneumoniae* (Kp) is a Gram-negative opportunistic bacterium, from the Enterobacteriaceae family, which classically produces lobar pneumonia, particularly in immunocompromised patients, and it is known to cause hospital-acquired pneumonia, meningitis, bloodstream, and urinary tract infections [1,2].

In recent decades, new challenges regarding Kp have emerged. One concerns the growing evidence of multidrug resistance strains [3]. Therapeutic management is threatened by the complex mechanisms of multidrug resistance, mainly due to extended-spectrum beta-lactamases (ESBL), AmpC beta-lactamases, or carbapenemases [4,5], of the strains, encountered in the hospital environment [6,7,8].

Another problem is the global spread of hypervirulent Kp (hvKp), an evolving pathotype, characterized by increased virulence that may cause disabling and life-threatening diseases in previously healthy individuals [6,7]. The first hypervirulent Kp endogenous endophthalmitis was reported in 1986 in Taiwan, in a patient with hepatic abscess without any underlying hepatobiliary pathology, raising attention about a new invasive strain [8,9,10]. Subsequent clinical trials reported that an epidemiologic shift in the epidemiology of cryptogenic liver abscess, with a growing number of case reports and subsequent clinical studies on larger patient groups signaling endemic growth of hvKp infections, accounting for 80% of cases in the Asian Pacific rim [11,12,13,14,15,16,17], and emerging in other regions such as the US, Australia, South Africa and recently in Europe [18,19,20,21,22,23,24,25,26,27], characterized by a high rate of metastatic septic dissemination, endophthalmitis and CNS complications being among the most common.

Hypervirulent Kp strains are recognized presently as the primary cause of endogenous endophthalmitis, whereas endogenous endophthalmitis is rarely seen with cKp, excepting on rare occasions in the setting of neutropenic or otherwise immunocompromised patients [28]. Moreover, the endogenous endophthalmitis in the clinical setting hepatic, or less frequently urinary, the pulmonary focus of infection is highly suggestive for hvKP [28,29,30,31].

### Particular Features of the Hypervirulent K. pneumoniae (hvKp) Strains

Most of the hypervirulent strains express a hypermucous phenotype, according to the string test: formation of mucoviscous strings greater than 5 mm, when a standard bacteriological loop is passed through a colony [32,33,34,35,36,37,38,39,40,41,42,43,44]. These strains are characterized by a super capsule, more resistant to complementary, neutrophil-mediated bactericidal activity and phagocytosis. While some authors contest the clinical significance of K2 serotype, K1 seems to more resistant to serum resistance assay and more than 40% of infected patients are free from diabetes or other comorbidities [35]. The K1 capsular polysaccharide is thought to be related with the resistance to phagocytosis, but the mechanism is not yet fully understood. A “Trojan horse” mechanism has been postulated with hvKp strains being able to survive and migrate within neutrophils and delay their apoptosis, within 24 h [6,36,37].

Another marker of virulence characteristic for hvKp is the magA gene, which is located in the K1 strain-specific wzy allele. The magA gene codes an outer membrane protein essential for the formation of a protective exopolysaccharide web associated with mucoviscosity and virulence of the K1 strain [38]. Two plasmid-encoded virulence factors have been well characterized, rmpA, a regulator of mucoid phenotype that upregulates capsule synthesis, and the iron siderophores (aerobactin, yersiniabactin, salmochelin) which enable the bacterium to obtain iron essential for growing, by chelating the Fe-binding proteins of the host [40,41,42,43,44,45]. Other virulence factors include the chromosome-encoded virulence genes kfu/PTS, which codes for an iron uptake system (kfu) and a phosphoenolpyruvate, and allS gene, which is associated with the anaerobic metabolism of allantoin [21,39,41].

HvKp strains identified in previously published paper present in most cases a natural “wild-type” resistance to antibiotic profile, suggestive for community-acquired infections, with uniform resistance to ampicillin and piperacillin and susceptibility to cephalosporins, fluoroquinolones, aminoglycosides, and carbapenems [10]. A small group of cases exhibits extended spectrum beta-lactamases (ESBL), up to 10 % in various studies. Recently, the emergence of carbapenem-resistant, hypermucoviscous *K. pneumoniae* strains is mentioned in a hospital outbreak in China [46,47,48,49,50], causing severe, untreatable infections in healthy individuals. These may occur by several mechanisms involving plasmids transfer: either a hvKp strain acquires an antimicrobial-resistant plasmid, or a cKp strain acquires the hvKp specific virulence plasmid or a hybrid plasmid that contains both virulence elements and antimicrobial resistance genes (Table 1) [6].

This review aims to document the endogenous hvKp endophthalmitis (EKE) in terms of epidemiology, risk factors, clinical features, therapeutic management, and outcomes. We performed a comprehensive review of literature on PubMed and additional databases (Science direct-Elsevier, Springer Nature, Web of Science), by the terms “hypervirulent *Klebsiella pneumoniae*” and “endogenous endophthalmitis”. Case reports and clinical studies in the English language, reporting clinical data, and therapeutic management were included. The elements taken into account for further analysis were: type of article (case report/ clinical study), length of the follow-up period (if existed), number of patients with EKE included; comorbidities; bilateral ocular involvement; associated septic disseminations; hvKp strain, if documented; ophthalmic findings; general and local treatment; outcomes in terms of preserved visual acuity.

The reports and clinical studies included in this review were comparable in terms of patient selection, methodology, and documentation of the results. However, due to the relatively new emergence of this *Klebsiella pneumoniae* pathologic association, we encountered significant differences regarding the documentation of cases, therapeutic approach—both general and local, ophthalmic treatment, according to the development of knowledge and therapeutic protocols.

## 2. Epidemiology of Endogenous hvKp Endophthalmitis (EKE)

Since the first case described in 1986, EKE has become currently the most frequent cause of endogenous endophthalmitis in Asian countries, with an incidence of up to 9%, and with an emergent trend in US, Australia, and European countries [22,54]. Epidemiological studies evidenced an incidence of 100 higher of metastatic spread in hvKp vs. other microorganisms. The incidence is higher in middle-aged adults, 50–60 years, with a slight male predominance and associated metabolic disorders, diabetes mellitus being a constant association.

Up to 80–90% of cases appear in the context of hepatic liver abscess. However, the incidence of EKE in patients with KPLA varied between 3.4% [17] and 12.6% [55], with a total of 95 cases of endophthalmitis in 1455 cases of KPLA (6.5%). One of the reasons for this wide variation may be the fact that the incidence was reported to all patients with a pyogenic liver abscess in some studies (19), while others analyzed only hypervirulent K1, K2 KPLA [56], as a different pathogenic entity. In a systematic review performed by Hussain et al., in 2020, on 15 studies, totaling 11,889 patients with KPLA, an incidence of 4.5% (217 cases) of EKE was found, with a 95% confidence interval 2.4% to 8.2%. Among infections with K1serotype KPLA, Wang et al. found a 19% incidence of metastatic ocular or central nervous system complications [30], and 84.2% of these patients developed irreversible catastrophic disabilities, including loss of vision, quadriplegia, paraparesis, and/or impairment of the higher cortical function [38,57]. This highlights the need for clinical awareness about the possibility of catastrophic septic ocular or CNS complications from KPLA in previously healthy individuals. Location in the right hepatic lobe, segment VII and VIII [27,58,59,60,61,62], size larger than 5 cm and thrombosis of the suprahepatic vein are associated with a higher risk of metastatic spread.

Although hvKp commonly causes a hepatic abscess, and endophthalmitis may occur in this clinical setting, endophthalmitis can occur from hvKp in conjunction with non-hepatic sites of infection, such as the urinary tract (5.5–8%) or lung infections (4–11%) [63,64]. Only one study [30] found an equal incidence of 40% each for liver and lungs as the primary focus of EKE. Post-spider bite cellulitis [65,66], prostate abscess, endocarditis [30], and osteomyelitis were reported as unusual causes of hvKp endophthalmitis,

The entry gate is considered to be the gastrointestinal tract [15], oropharyngeal [67], or recent urinary tract infections [62]. Oropharyngeal colonization could lead to infection with aspiration, but it is still unclear what the mechanism of entry is via the GI tract since many individuals subsequently infected have no apparent GI pathology. Entry through micro-breaks in the skin, similar to *Staphylococcus aureus* is also a theoretical possibility. However, the exact mechanism and necessary time lapsed between inoculation and the onset of the invasive systemic infection is not fully understood [41].

Chung et al. found a significant fecal carriage of K1 and K2 pathogenic stains in stools in a cohort of healthy South Korean residents [15]. Fujita et al. found a correlation with previous treatments with ampicillin or amoxicillin, leading to the hypothesis that these antibiotics may increase the risk for KPLA by selection pressure [67,68,69]. Diabetes, by impaired gastrointestinal permeability, may favor bacterial translocation into the portal bloodstream [68,69]. In a two-case report, Fujita et al. found KPLA and endophthalmitis appeared in patients previously incompletely treated with amoxicillin/clavulanic acid for sore throats and upper respiratory infections and postulated the possibility of systemic dissemination following incomplete eradication of the pathogen [67].

## 3. Risk Factors

This emerging epidemiological issue was initially attributed to the association with the Asian race and diabetes [10,27,36,59,60]. However, a significant number of published papers documented the presence of the infectious strain in young, previously healthy subjects, of all ethnic groups [18,19,22,25]. In recent years, cases from United States, Belgium, Spain, UK, Germany, Sweden, Romania, Saudi Arabia have been reported, indicating global dissemination of these strains causing invasive liver abscesses and endogenous endophthalmitis in both Asian and non-Asian patients, independent from their immunocompetent status or coexistence, or coexistence with diabetes [70].

Diabetes was frequent comorbidity [18,25,26,71] with an incidence between 35–92%, although it may appear in previously healthy adults (Table 2 and Table 3). The role of poor glycemic control in the impaired phagocytosis of capsular serotypes K1 or K2 of *Klebsiella pneumoniae* was postulated for the pathogenesis of serious metastatic complications in diabetic patients [72], but the hypothesis was not supported by further experimental studies [73]. On the other hand, Coburn et al. demonstrate that a possible explanation of the frequent association of diabetes with EKE following KPLA may be the specific diabetic ocular environment, with increased retinal blood barrier permeability [74]. Another possible explanation may be the disturbing gastrointestinal defense mechanisms in diabetes, which may favor the bacterial translocation from the gastrointestinal lumen into the bloodstream [15].

Chung et al. found also an increased incidence of other metabolic disorders, such as hypertension and fatty liver [15], but it is not clear what specific mechanisms may favor the invasive infection in these cases.

Other associated pathologies were: thrombocytopenia [12], anti-interferon type 3 (anti-IFN3) autoantibodies [24], hypertension [26,55], and less frequent cirrhosis and Guillain Barre syndrome [17]. IFN-3 is considered to be involved in the neutrophil response, hepatic macrophage activity, and mucosal immunity. IFN-3 signifies type 3 interferon, which is considered to be involved in the neutrophil response, hepatic macrophage activity, and mucosal immunity. The identification of the anti-IFN3 in the case reported by Castle et al. may explain the poor outcome, with bilateral vision loss, in a previously healthy patient [24].

## 4. Diagnostic of Endogenous hvKp Endophthalmitis

The distinctive clinical course of irreversible ocular injury caused by hvKp strains is still not complete elucidated. However, experimental animal studies showed a significant decrease in electric activity of the infected retina within 18 h from the onset [34].

In 25.8% (5–50%) of cases, the septic involvement was bilateral, resulting in permanent vision loss. Bilaterality was encountered in 5–50% of cases in the reviewed studies, in association with multiple septic determinations, such as lungs [38], meninges and CNS, coagulopathy, thrombocytopenia, urinary sepsis (Table 2 and Table 3). There were a total of 78 bilateral cases in a total of 302 patients with EKE and KPLA (25.8%). In all cases with ocular determination, the localization of liver abscess was in the right hepatic lobe (Figure 1 and Figure 2)

### 4.1. Clinical Evaluation. Prognostic Factors

Various reports found that the ophthalmologic findings, with decreased vision and painful eye, may precede the signs related to the initial focus of infection [10,55,56,62,67]. The most common presentation was blurred vision and ocular pain. In the study of Park et al., ocular symptoms developed prior to the diagnosis of liver abscesses in 66.7% of cases [15]. These findings were confirmed by other case reports [26,67]. General signs are suggestive of an infectious syndrome, although the fever may not be present at admission.

Ocular findings included chemosis, red painful eye, haze cornea, and vitreous. Decreased visual acuity at presentation of hand movement or less and hypopyon were associated with poor prognosis [15,68,72,75,76,77]. Patients with unilocular involvement were found to be a risk factor for evisceration, in comparison with bilateral infection, one possible explanation being the delay in presentation in cases in which only one eye was affected [16]. In a study by Ang et al [16], early onset of the ocular symptoms was also a factor of poor prognosis, in correlation with the virulence of the pathogen and bacterial load.

Hypopion may be present in 30–40% of cases and is considered a factor for adverse outcomes [15]. EKE is the result of the metastatic septic emboli in the choroid, with subsequent development of the infection through a blood–retina barrier, into the retina and vitreous body. Passage of the germs, leukocytes, and inflammatory products into the anterior segment of the eye may occur at the level of zonular fibers. The increased number of cellularity and fibrin at the level of the anterior chamber is an expression of intense inflammation in the posterior segment [15].

According to the focal or diffuse aspect, as well as the extension of the septic process, various clinical forms were described: subretinal abscess, posterior diffuse involvement, anophthalmia, scleral abscess with spontaneous perforation, orbital cellulitis.

### 4.2. Imagistic and Laboratory Tests

Ocular imagistic investigations (ultrasonography) CT and an IRM exam may indicate the increased density of vitreous body +/− subretinal abscess, as well as the extension of the septic process with eyeball disorganization and infiltration of the orbital soft tissue. Documenting the initial focus of infection and the possible septic associations requires hepatic, renal pulmonary, and CNS imaging, or other locations suggested by clinical findings.

Aqueous humor and vitreous taps may identify the pathogen at the eye level in 30–40% of cases. Additionally, *K. pneumoniae* was isolated from the blood culture in most cases. A string test is an inexpensive tool, which can strongly suggest the involvement of a hypervirulent strain of *Klebsiella pneumoniae*. However, physicians should be aware that a negative string test does not exclude the presence of hvKp in cases with the clinically documented invasive syndrome. Extensive documentation of the serotype and virulence gene by genomic sequences and PCR was performed in recent studies and reports, revealing a high incidence of K1 serotype and magA gene [5,21,29].

## 5. Management of EKE

### 5.1. General Antibiotic Therapy and Assesment of the Primary Focus

The antibiotic regimen should be chosen considering the penetration of the available antibiotics in different tissues, like the eye and CNS, when the metastatic spread is encountered. In particular, third-generation cephalosporin is recommended due to its good penetration in the vitreous cavity and cerebral spinal fluid. Peak vitreous concentrations of at least 2 mg/l can be achieved [56]. The clinical management was based on 2–3 weeks intravenous antibiotic therapy with the 3rd or 4th generations of cephalosporins, which are considered to be superior to the first generation of cephalosporin in septic metastasis and recurrence prevention [15,16,18,19,37,68]. In some papers, metronidazole or piperacillin/tazobactam was added for anaerobic pathogens [20,23]. The use of aminoglycosides is debatable, due to the relatively low penetrance of the abscess capsule. However, they may be useful upon pathogens in the bloodstream, thus preventing septic dissemination [10,12,50,68]. After 2–3 weeks, if the criteria for favorable evolution are met, the patients were switched to oral fluoroquinolones, for 1–2 months, with periodic follow-up to identify possible recurrence early [10,23,26].

ESBL strains were identified in up to 10–13% of cases [40,57,58,61]. In a recent study on 110 patients with KPLA, with a mean follow-up period of 3.65 years, Wang et al. [76] found that ESBL was encountered significantly more frequently in the patients who experienced recurrence (30.0% vs. 8.89%). This finding supports the idea that ESBL should be considered an independent risk factor for the recurrence of KPLA [76]. and in these cases, cephalosporins should be replaced with imipenem or meropenem therapy [21]. In the absence of ESBL, such use of reserve antibiotics should be avoided, to preserve as long as possible the natural “wild-type” phenotype of hypervirulent KP strains.

In non-responding cases, percutaneous or surgical drainage of the primary infection focus should be performed to mitigate the hematogenous spread. Percutaneous drainage of the liver abscess, ultrasound or preferable CT guided, may be performed either needle assisted or by continuous catheter drainage with negative pressure. The indications for drainage include febrile patient after more than 48–72 h of adequate intravenous antibiotherapy, abscess of more than 6 cm in diameter, or impending perforation [50,78,79]. Multiple location and multiloculated abscesses are relative contraindications due to increased chances of failure. High locations may not be suited for guided drainage due to the increased risk of pneumoperitoneum. When a catheter is used, one must take into account the specific high viscosity of the KP strains, thus larger size and frequent flushing should be used to prevent blockage [68]. The catheter was usually maintained for 1–2 weeks and removed when drainage is sterile, less than 5 mL daily, and fever does not reappear when the tube is clamped [57].

Surgical drainage should be considered if a patient fails to improve at a satisfactory rate with percutaneous drainage and intravenous antibiotics, in cases of impossible percutaneous evacuation due to thick pus, multiloculated or multiple liver abscesses, located in the left hepatic lobe. Surgery is mandatory in cases of spontaneous rupture or if other biliary pathology should be addressed. Abscess spontaneous rupture is a serious complication, requiring immediate surgery. It is favored by: dimensions of more than 5 cm, thinned walls, intracavitary gas formation, and hyperglycemic dysregulation in diabetic patients [68].

### 5.2. Ocular Management and Visual Outcomes

Early case reports and clinical studies using topical antibiotics, corticoids, and mydriatics reported a poor outcome resulting in most cases in blindness and anophthalmia due to spontaneous eye perforation that required evisceration or enucleation [18,20,60,61].

#### 5.2.1. Intravitreal Antibiotics

Today, the mainstay of ocular management is intravitreal antibiotics based on a combination of 3rd-generation cephalosporins +/− vancomycin +/− aminoglycoside. The role of intravitreal antibiotics is supported by evidence that vitreous samples are still positive for *Klebsiella*, despite intensive systemic antibiotherapy. According to recent guidelines of endophthalmitis treatment, intravitreal antibiotics increase the third time chance to preserve some degree of vision in the affected eye and decreased by a third the risk for evisceration. 

Intravitreal drugs used were: ceftazidime 2.25 mg/0.1 mL +/− vancomycin 1–2 mg/0.1 mL+/− amikacin 0.4 mg/0.1 mL [13,15,16,18,29,55]. There are no current guidelines regarding the number and frequency of intravitreal antibiotics administration, the therapeutical decision is based on the clinical response. The median number of administrations in the reviewed studies was 3–5 [12,58,67,77]. However, Martel successfully treated a Kp subretinal abscess with intensive intravitreal ceftazidime and vancomycin, daily for one week, then at 48 h, with a total of 13 administrations [80].

#### 5.2.2. Intravitreal Steroids

There is no consensus regarding the utility of intravitreal triamcinolone in EKE. Intravitreal corticoids administration was found to be associated with vision gain of more than 6/60 [22,55] and may be performed after 2–3 intravitreal antibiotic injections. However, in the reviewed studies, corticoids were administrated in only 20–50% of cases.

#### 5.2.3. Pars Plana Vitrectomy (PPV) and Subretinal Abscess Drainage

The benefits of early pars plana vitrectomy are still a subject of debate in the clinical management of EKE. The timing and indications for PPV are a challenging issue, with contradictory results. Some authors recommend early vitrectomy, considering as beneficial results the decrease of intraocular bacterial load and increased penetration of the antibiotics into the eye, preventing irremediable damage of the eye structures. Connel et al. reported a 0% evisceration and enucleation rate in eyes treated by PPV and intravitreal antibiotics when compared to 50% in eyes treated with intravitreal antibiotics only [79]. However, Ang et al. found that early vitrectomy is not associated with a better visual outcome [16]. Martel et al. raised concern about the efficacy and results of PPV on an inflamed necrotic retina [80]. A vitrectomy could result in supplementary damage of the retina, especially if the vitreous body is not detached due to vitreoretinal tractions. Another aspect to be taken into account is that removing the vitreous body will reduce the time of contact with the antibiotic with the infected retina. An absolute contraindication for PPV is the presence of life-threatening septic dissemination, such as meningitis or CNS abscess. Early removal of vitreous abscess can favor rapid evacuation of the pathogen load and mediators of inflammations. A more conservative approach would indicate PPV only in eyes with VA severely decreasing a worsening grade of relative afferent pupillary defect with no response to intravitreal antibiotics for 48 h [18,80]. Choosing the surgical timing should take into account the general status of the patient, hemodynamics, and coagulation parameters, which can be affected during *Klebsiella* infection.

In cases with abscesses larger than four disks areas, retinotomy and abscess drainage may be an option, but the risk for post-operative retinal detachment and vitreoretinal proliferation is higher due to active inflammation, laser photocoagulation is difficult on a fragile retina. Venkatesh used a conservative approach, by combining PPV with 41G needle abscess drainage, which has the advantage of an easily self-sealing retinal hole [81].

PPV alone may be also beneficial even in large-size retinal abscesses according to Xu et al. [29], by removal of the toxins and inflammatory products. 

#### 5.2.4. Lensectomy

Phacoemulsification of the lens was employed by some authors, but with limited results in vision improvement and the risk of corneal melting and spontaneous perforation [38].

#### 5.2.5. Evisceration and Enucleation

Evisceration or enucleation was performed in cases non-responsive to other therapeutical options, with spontaneous perforation and total vision loss. 

Despite immediate management, the visual outcome in patients with EKE is poor [82,83]. It ranges from hand motion visual acuity to evisceration or enucleation of the eye in more than 75% of cases [74,82]. In a study of Ang et al., on 71 eyes of 61 patients with EKE, the severe visual loss was encountered in 76% of cases, despite adequate treatment. with a high likelihood of bilateral blindness [16]. However, early intervention in the first 48 h from the onset, with intravenous 3rd or 4th generations of cephalosporins, intravitreal antibiotics +/− corticosteroids and, in severe, non-responsive cases, pars plana vitrectomy, may preserve a vision of more than 6/60 in 13.6% of cases [13]. Several reports showed that early treatment before severely decreased vision appears, based on systemic and intravitreal antibiotics may preserve vision in variable degrees [13,16,45]. The introduction of an ophthalmological screening program for patients with *KPLA* was useful in early diagnosis and reducing the risk of progression to enucleation of eyes due to earlier diagnosis and more aggressive treatment [16,72,75,84].

## 6. Conclusions

Global dissemination of hvKp is a major epidemiologic shift that should be considered in the diagnostic and therapeutic management of patients with endogenous endophthalmitis. There is a concern regarding the possibility of abelon further combination of virulence and resistance, causing severe, untreatable infections in healthy individuals, which would be extremely difficult to manage [34,73]. Ophthalmologic screening in patients with KPLA or other suspected primary hvKp infections is extremely important for early diagnosis that could preserve visual function. Once vision is severely decreased, the outcome of hvKP endophthalmitis is extremely severe, despite aggressive ophthalmological treatment. The dramatic evolution of the ocular damage, with loss of vision in most cases and anophthalmia in 25–50% of cases, requires awareness of this new pathological association and the need to follow up these patients in multidisciplinary teams and to initiate aggressive ophthalmic treatment early concomitantly with antibiotic treatment as this is essential for saving vision.

## Figures and Tables

**Figure 1 life-11-00676-f001:**
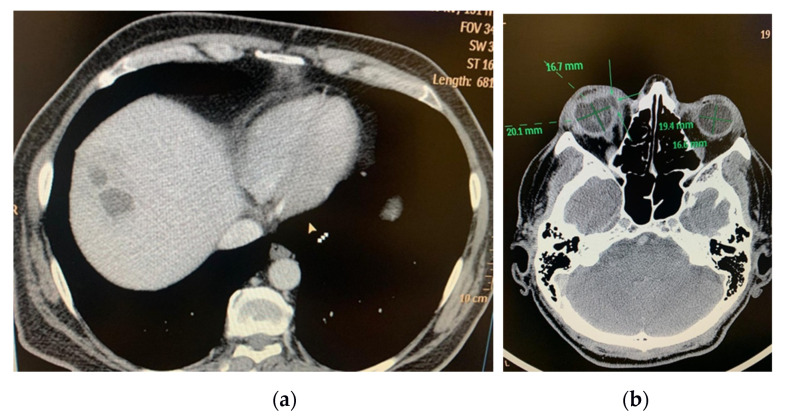
(**a**) CT exam: multiloculated hepatic liver abscess, right lobe, segment VIII, 50/32/57 mm; (**b**) CT Exam: Right eye endophthalmitis, eyeball with irregular contour, with a hypodense area in the internal side (at the area of the insertion of the medial rectus muscle), with possible communication between posterior chamber and periorbital space (perforation) (archive of the Ophthalmology Department, Emergency University Hospital Bucharest).

**Figure 2 life-11-00676-f002:**
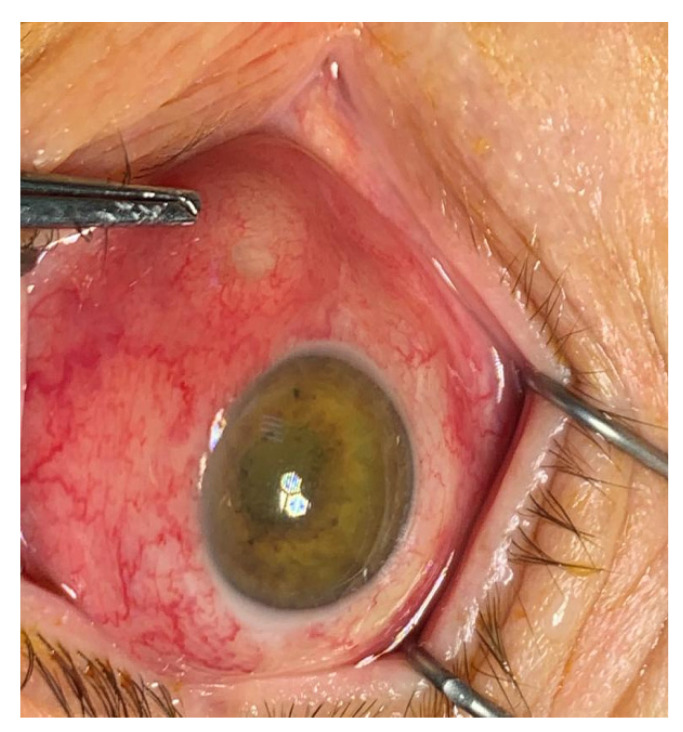
Endogenous hvKp endophthalmitis (EKE): corneal haze, intense hyperemia, scleral abscess in the internal angle, with impending ocular perforation (archive of the Ophthalmology Department, Emergency University Hospital Bucharest).

**Table 1 life-11-00676-t001:** Differences between common *K. pneumoniae* (cKp) and hypervirulent *K. pneumoniae* (hvKp).

	cKp	hvKp
Acquisition of infection	mostly hospital acquired	mostly community acquired
Host status	immunocompromised, old, hospitalized	previously healthy, all ages;
Geographic region	globally spread	first endemic in South East Asia, with progressive global spreading [51]
Locus of primary infection	lungs, urinary tract, blood stream, meningitis	liver, less frequently kidney, lungs, others
Co-pathogens	unfrequently; mostly monomicrobial	may be plurimicrobial, especially is digestive, hepatobiliary, urinary or soft tissue infections [6,51]
Metastatic spread	unfrequently	Frequent: endophthalmitis, meningitis, central nervous system (CNS), psoas, prostatic abscess, necrotizing fasciitis) [51,52]
Phenotype	String < 5 mm	hypermucoviscosity, string > 5 mm
Serotypes	K1–79	mostly K1, K2; other serotypes identified: K5, K16, K20, K54, K57, KN1
Multidrug resistance (MDR)	Frequent, especially in hospital acquired infections (to cephalosporins, fluoroquinolones, carbapenems)	less than 10–15% (but on a emerging trend)
Siderophores	enterobactin, yersiniabactin [6,53]	enterobactin, aerobactin, yersiniabactin, salmochelin [6,53]

**Table 2 life-11-00676-t002:** Demographics, clinical characteristics and treatment of patients with *Klebsiella pneumoniae* endogenous endophthalmitis (EKE) from case reports.

Author, Country, Year	Sex Age	Underlying Diseases	Serotype	Focus of Infection	Ophthalmic Findings	Associated Septic Spread	General Treatment	Opththalmic Treatment	Outcome
Seo, R^58^ Japan, 2016	F, 64	-	K2A, rmpA positive	KPLA ^a^ right lobe	VA ^b^: NLP ^c^ Chemosis, corneal, vitreal haze	thrombocytopenia, coagulopathy	iv ^d^ meropenem (3 g/day), changed to ceftriaxone (2 g/day) percutaneous drainage of KPLA	Early VIT ^e^	VA: NLP
Saccente M^18^, US, 1999	M, 38	diabetes	No info	KPLA right lobe,	VA: 20/400, Hypopyon, chemosis, vitreal haze	meningitis	iv ceftriaxone (21 days), metronidazole (17 days), followed by oral levofloxacin and metronidazole, 30 days	Intravitreal ceftazidime and vancomycin; Topic ciprofloxacin, cefazolin and corticosteroids 1 drop/1–2 h	VA: 20/60
Maruno T^38^, Japan, 2013	M, 63	-	K1	KPLA right lobe,	VABE: LP ^f^, bilateral endophthalmitis and orbital cellulitis	lungs septic emboli	iv meropenem 1 g/kg every 8 h, changed to ceftriaxone Ultrasound-guided percutaneous transhepatic liver abscess drainage	No info	Bilateral total vision loss
Sobirk S^20^, Sweden, 2010	M,45	-	Mucoid strain	KPLA multiloculated	VA: LP; Chemosis, vitreous hemorrhage	-	iv ceftazidime and vancomycin	Topic antibiotics	Vision loss
Baekby M^21^, Denmark, 2018	M, 78	-	K1, hv CC23 clone	multiloculated KPLA > 12 cm	VA: LP, red painful eye	lumbar abscess	iv piperacillin/ tazobactam, followed by oral ciprofloxacin	Intravitreal ceftazidime and vancomycin	Vision loss
Fujita M^38^, Japan 2015	F, 70; M, 50	-	K1	KPLA	Case1: VARE: LP; hypopyon, haze of ocular media Case 2: VARE ^b^*: LP; hyperemia, corneal and vitreous haze,		case 1: iv sulbactam /cefoperazone; percutaneous drainage Case 2: iv clindamycin and cefazolin, then switch to cefepime	Case 1: VIT + PEA ^g^+ antibiotics Case 2: VIT+ PEA, per secundam IOL ^h^, antibiotics	Case 1: Corneal melting; evisceration Case 2: preserved partial vision
Al Mahmood^71^, Saudi Arabia, 2011	M, 43; F, 70	diabetes	No info	KPLA	Case 1: VA: LP, hemorrhagic purulent vitreous Case 2: VA BE **: LP; ocular hypertension, hypopyon		iv ceftazidime vancomycin	Case1: vancomycin 1 mg/0.1 mL, ceftazidime 2.25 mg/ 0.1 mL, amphotericin B 10 lg/0.1 mL and dexamethasone 0.4 mg/0.1 mL Case 2: + VIT	Case 1: VA: NLP Case 2:RE: evisceration; VA LE ***: LP
Wells JT^23^, US, 2015	F, 67	-	K1	KPLA	RE: pan ophthalmitis, retinal detachment	peritoneal sepsis, septic shock	iv ceftriaxone, changed to imipenem, vancomycin, and fluconazole	Intravitreal Vancomycin and ceftazidime	deceased
Castle G^24^, UK, 2020	M, 56	anti-IFN-3 ^i^ autoimmunity	No info	KPLA 9 cm, multiloculated	VARE:1; choroidal abscess VA in LE: HM, marked vitreous and anterior chamber haze	lung emboli	iv ceftriaxone, followed by oral ciprofloxacin	RE: Vitrectomy, lens removal, intravitreal antibiotics LE: evisceration	RE: LP, aphakic LE: NLP
Paraschiv F^25^, Romania, 2018	F, 53	diabetes	positive string test	KPLA	VA CF Uveitis, retinal hemorrhages	meningitis thrombocytopenia	iv vancomycin and ceftriaxone, percutaneous drainage	Intravitreal and topic antibiotics, topic corticosteroids	Partial recovered vision
Van Keer J^26^, Belgium, 2017	M, 84	hypertension	No info	KPLA < 30 mm	VARE 20/250; proptosis, chemosis VALE: CF	-	iv ceftriaxone	Intravitreal ceftazidime, topic antibiotics and cs; LE: vitrectomy	RE: LP LE: partial vision recovery
Pichler C^5^, Germany, 2017	M, 61	-	K2 serotype ST2398	KPLA	endophthalmitis	thrombosis of supra hepatic vein	percutaneous catheter drainage, i.v. piperacillin/tazobactam, and ciprofloxacin	No info	No info
Xu ^29^, China, 2019	M, 25	diabetes	KP587, ESBL ^j^	KPLA	VA: LP, hypopyon		i.v. imipenem	Intravitreal imipenem, vancomycin and dexamethasone, VIT	CF ^k^ at 0.4 m
Abdul Hamid, A^63^, UK, 2013	M, 36		No info	KPLA, 6 cm, segment VIII	VA: LP, chemosis, sever periorbital edema	tenosynovitis, urinary infection	oral ciprofloxacin, 10 weeks	Intravitreal amikacin 0.4 mg/0.1 mL and vancomycin 1 mg/0.1 mL topic antibiotics, CS, mydriatics	VA: NLP
Sridhar J^65^, 2014, US	M,43 F, 58 F,60	DM DM Multiple mieloma	No info	KLPA Spider bite + celullitis unknown	VA: 2/200 VA:1/200 VA: LP	no info	no info	Vitreous tap+ Intravitreal vancomycin and ceftazidime − all patients + intravitreal CS − patient 1 + VIT − patient 2	Enucleation Evisceration enucleation
Hassanin F^75^, 2021, Saudi Arabia	F, 55	DM	+ string test	Renal abscess	VA: LP, Chemosis, conjunctival injection, hypopion, panopthalmia with orbital cellulitis	-	iv ceftazidime, vancomycin, and metronidazole	VIT+ intravitreal vancomycin and ceftazidime evisceration	Anophthalmia
Dubey D^66^, 20313, US	F, 41	DM	K1, sensitive to carbapemens only	Renal abscess	blurred vision, hypopion, endophthalmitis	lungs, CNS ^l^	iv meropenem (2 g /8 h), 8 weeks	Intravitreal vancomycin and ceftazidime	No info
Martel A^76^, 2017	M, 60	-		KPLA	VA 20/50 Anterior uveitis, subretinal abscess	urinary	iv ceftazidime	Intravitreal ceftazidime (2.25 mg/0.1 mL), 13 injections	VA 20/20

^a^ KPLA: *K. pneumoniae* liver abscess; ^b^ VA: visual acuity, * RE: right eye; ** BE: both eyes; *** LE: left eye; ^c^ NLP: no light perception;^d^ iv: intravenous; ^e^ VIT: pars plana vitrectomy; ^f^ LP: light perception; ^g^ PEA: phacoemulsification of the lens; ^h^ IOL: intraocular lens; ^i^ anti IFN-3: anti-interferon 3; ^j^ ESBL: extended spectrum beta lactamase; ^k^ CF: counting fingers; ^l^ CNS: central nervous system.

**Table 3 life-11-00676-t003:** *Klebsiella pneumoniae* endogenous endophthalmitis (EKE) in clinical studies.

Author, Country, Year	Study Period	No of Cases/No of Eyes	Sex (M/F)	Age (Mean)	Comorbidities	K.p. Strain	Focus of Infection (%)	Other Disseminations	Ocular Outcome	General Therapy	Ophthalmic Therapy
Chiu CT^59^, Taiwan, 1988	1977–1986	3(5.2% incidence in KPLA)	2/1	65	diabetes (66.7%)	no info	hepatic	-	vision loss (3) enucleation (1)	iv cephalosporins	topic antibiotics, mydriatics, CS ^a^
Liao HR^60^, Taiwan, 1992	1983–1988	12 (25% bilateral)	no info	no info	diabetes (91.66%)	no info	hepatic (50%) Others (urinar, pulmonary)	-	no light perception (9); enucleation or evisceration (6) light perception (3)	iv cephalosporins percutaneous drainage of KPLA if necessary	topic cephalosporin + gentamicin, mydriatics, CS
Cheng DL^61^, Taiwan, 1991	1981–1987	14; (14,2% bilateral)	no info	no info	diabetes (50%)	No info	hepatic	lungs (4) CNS (3) prostate (1)	12 -blindness 2—partial blindness	general	Topic antibiotics
Fung C^10^, Taiwan, 2002	1991–1998	14 (10.44%); 134	3/1	56.4	diabetes (93%)	K1 (85.7%) K2 (14.3%)	hepatic	lungs (3) CNS (2)	deceased (4) vision loss/very low vision (8) vision recovery (2)	pigtail catheter drainage; iv 3rd generation cephalosporin + gentamicin, 2 weeks, continued with oral ciprofloxacin	No info
Fang CT^56^, Taiwan, 2007	1997–2005	14 (12.6%); 177 28.5% bilateral	8/6	58.2	diabetes (93%; 78.4%)	K1 (92,85%) K2 (7.15%)	hepatic	meningitis (8) spondylitis/diskitis (4) pneumonia (1) brain abscess (2) fasciitis (1)	vision loss (8; 57.1%) limited vison (2; 14.2%) partial recovery (4;28.5%)	pigtail catheter drainage; iv third generation cephalosporin + gentamicin	Intravitreal antibiotics and CS
Yang CS^55^, Taiwan, 2007	1994–2001	22 (bilateral 22.7%)	19/3	54.6	diabetes (68%)	no info	hepatic	lungs (6) CNS (3) kidney (1) prostate (1)	vision loss (89%), of which 41% anophthalmia VA > 1/10 (3 cases)	iv cephalosporins and aminoglycosides	intravitreal antibiotic/CS enucleation/evisceration 41%
Sheu SJ^12^, Taiwan, 2011	1991–2009	42(6.9%); 602 (26% bilateral)	26/16	60.2	diabetes (35%)	no info	hepatic	no info	VA of CF or better in 35.8%	iv cephalosporins, intravitreal antibiotics; VIT (9 cases)	intravitreal ATB ^b^ -40 (amikacin 400 mg/ 0.1 mL, gentamicin 0.05 mg/0.1 mL, ceftazidime 2.25 mg/0.1 mL, and vancomycin 1 mg/0.1 mL intravitreal CS -12 VIT-9 enucleation/evisceration- 11
Park IH^13^, South Korea, 2015	2004–2013	12	7/5	64.3	diabetes (50%; 23.8%)	no info	hepatic (incidence in KPLA 6.1%)	no info	NLP (7) HM ^c^ (3) Partial recovery (2)	iv cephalosporins and aminoglycosides	intravitreal ceftazidime 2.25 mg/0.1 mL and vancomycin 1 mg/0.1 mL; early VIT
Lee JY ^14^, South Korea, 2014	1997–2013	8;	5/3	71.1	diabetes (50%)	no info	hepatic	no info	vision loss (7) partial recovery (1)	iv ceftriaxone,	intravitreal antibiotics (7)
Pastagia M^19^, US, 2008	2001–2007	1 (bilateral EKE)	no info	no info	diabetes (100%)	K1	hepatic	lungs, meninx (1)	bilateral vision loss	iv ceftriaxone /imipenem in cases with ESBL Intravitreal antibiotics	no info
Chung CY^15^, Hong Kong, 2016	2006–2015	19 (bilateral 26.3%)	12/7	67.89	diabetes (11) hypertension (5) cirrhosis (2) Guillain-Barre (1)	no info	liver 18 (94%) urinary 1 (6%)	coagulopathy (1), shock (1) psoas (1) CNS (1) pleura (3)	deceased (4) anophthalmia (9) total vision loss (3) VA > 0.3 (3)	iv cefuroxime, gentamicin;	intravitreal ceftazidime and amikacin+/− vancomycin 17/19 (89.47%) VIT 3 (15.8%) evisceration 9 (47.4%)
Ang M^16^, Singapore, 2011	1986–2007	61/71 (18% bilateral)	49/12	55.7	diabetes (55.7%)	no info	liver 46 (77.5%) urinary 11 (9.9%) repiratory 3 (4.2) other 6 (8.4%)	lungs (5) meningitis (5) urinary (7)	evisceration (19;26.8%) VA < 20/400 55 (77.5%) VA > 20/400 16 (22.5%)	iv ceftriaxone,	intravitreal vancomycin (2.0 mg/0.1 mL) plus ceftazidime (2.25 mg/0.1 mL) or amikacin (0.4 mg/0.1 mL).
Tan YM^17^, Singapore, 2003	1995–2001	10 (3.4%); 289	6/10	45.7	diabetes 70%	no info	hepatic	skin emboli (1) pneumonia (1) ARDS ^d^ (2) pleural effusion (1)	evisceration (2) vision loss (4) poor vision -HM (4)	iv ceftriaxone and gentamicin/metronidazole	Intravitreal antibiotics, VIT
Odouard C^22^, Australia	2011–2015	4 (50% bilateral)	3/1	39.2	diabetes (50%)	No info	hepatic	lungs (1) prostate (1)	enucleation (1) vision loss (1) preserved vision >6/24 (2)	iv ceftriaxone, hepatic abscess drainage	intravitreal ceftazidime and vancomycin, VIT
Shields RA^30^, 2017, US	2000–2017	10/12 (20% bilateral)	8/2	56	diabetes 7 (70%)		liver 4 (40%) lungs 4 (40%) endocarditis 1 (10%) osteomyelitis 1 (10%)	no info	anophthalmia 5/12 vision loss 2/12 VA > 20/300—5/12	iv antibiotics	intravitreal antibiotics (1–33 injections)–10/12 enucleation, evisceration 5/12
Lim H^31^, 2014, S. Korea	2005–2011	18/23 (27% bilateral)	14/4	68.7	diabetes-8 (44%) cirrhosis 4 (22%) psoriasis arthritis 1(5%)	no info	KPLA 15 (85%) lungs (11%) prostate abscess 1(5%)	no info	anophthalmia 1 (2.3%) total vision loss 13 (56%) VA> 0.05- 9 (39%)	iv ceftazidime/ceftriaxone+ amikacin/metronidazole/tazocin /meropenem	intravitreal vancomycin, ceftazidime +/− gentamicin/amikacin VIT- 7 (30%) evisceration 1 (2.3%)
Chen SC^84^, 2016, Taiwan	2002–2013	48/58 (20.8% bilateral)	29/19	59.3	diabetes 34 (70.8%) hypertension 17 (35.4%) malignancy 7 (14.6%) cirrhosis 4 (8.3%)	no info	KPLA 33 (68.7%) Urinary 4 (8.3%) pneumonia 2 (4%) other 5 (10.4%)	no info	anophthalmia 12 (20.6%) total vision loss 27 (46.5%) VA > 1/60 -19 (32%)	iv antibiotics, according to antibiogram	intravitreal ceftazidime+/− vancomycin (3.8 + −2.6) – 100% intravitreal CS -27 (50%) VIT 18 (31%) evisceration/enucleation 12 (20.6%)
Wong JS^64^, 2000	1986–1998	18/20 (11% bilateral)	10/8	49.3	KPLA 15 (83%) pneumonia 2 (11.1%) urinary 1 (5.5%)	No info	KPLA 15 (83%) pneumonia 2 (11%) urinary 1 (5%)	no info	vision loss 15 (83%) AV > 0.05 3(17%)	ceftriaxone+ imipenem/gentamicin/metronidazole	intravitreal cefazolin/ceftazidime + vancomycin/gentamicin -100%

^a^ CS: corticosteroids; ^b^ ATB: antibiotics; ^c^ HM: hand movement; ^d^ ARDS: acute respiratory distress syndrome.

## Data Availability

Not applicable.

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
