# Peer review of "Hypervirulent Klebsiella pneumoniae Endogenous Endophthalmitis—A Global Emerging Disease"

_life, 2021, doi:10.3390/life11070676_

Round 1
Reviewer 1 Report
The objective of this manuscript was to review hepatic abscess due to Klebsiella pneumoniae with which endophthalmitis was also present. The epidemiology and risk factors, microbiologic characterization of the hypervirulent K. pneumoniae (hvKp) strains, antimicrobial susceptibility, features of hepatic abscess and management, and endogenous endophthalmitis were discussed. This review summarizes and archives some important information. However, there are some conceptual and other issues that need to be addressed.
Major issues
- A major issue with this review is that it only addresses/reviews the combination of hepatic abscess and endophthalmitis. It would be more appropriate to frame this review as endophthalmitis that occurs in the setting of hvKp infection. Although hvKp commonly causes hepatic abscess, and endophthalmitis may occur in this clinical setting, endophthalmitis can occur from hvKp in the absence of hepatic abscess alone or in conjunction with non-hepatic sites of infection. As a result, by linking endophthalmitis with hepatic abscess the authors are not capturing the full clinical spectrum of disease. A search for papers that report on hvKp and endophthalmitis and variations thereof would be a more appropriate starting point.
- Introduction. The concept that K. pneumoniae has evolved into two pathotypes, hvKp and classical K. pneumoniae (cKp) should be introduced at the start. Some discussion on the epidemiologic, genomic, and clinical differences between these pathotypes should be included. hvKp strains are the primary cause of endogenous endophthalmitis, in fact this syndrome strongly suggests hvKp infection. Whereas endogenous endophthalmitis is rarely seen with cKp, excepting on rare occasions in the setting of neutropenic or otherwise immunocompromised patients.
- Section 3.1-antibiotic susceptibility. Some discussion of extensively drug resistant hvKp strains is in order. These occur by several mechanisms; a. A hvKp strain acquires an antimicrobial resistant plasmid, b. A cKp strain acquires the hvKp specific virulence plasmid, or c. A cKp strain acquires a hybrid plasmid that contains both virulence elements and antimicrobial resistance genes.
- A review of hvKp and hepatic abscess is not particularly novel. This review would be best if it focused on hvKp and endophthalmitis. There are a number of unresolved issues with endophthalmitis, especially with regards to management. So, I would delete sections 3.2 and 3.3.
- Section 3.4 should be deleted. This syndrome occurs routinely with hvKp infection, it is not a distinct subset of hvKp infections, and all hvKp infections are invasive.
- Section 3.5 should be greatly expanded. I would break it down into several section such as epidemiology, risk factors, clinical presentation, and most importantly management. This syndrome is largely hvKp specific, not K1 and K2 capsule type specific; this requires clarification. hvKp strains that are non-K1/K2 can cause this clinical syndrome.
Minor issues.
- Lines 183-189. Please don’t confuse colonization with entry for the reader. Colonization appears to be the first step requisite for infection, but colonization doesn’t necessarily lead to infection. Oropharyngeal colonization could lead to infection with aspiration, but it is still unclear what the mechanism of entry is via the GI tract since many individuals subsequently infected have no apparent GI pathology. Entry through micro-breaks in the skin, similar to Staphylococcus aureus is also a theoretical possibility.
- lines 199-200. The sensitivity and specificity of the string test is not optimal. A positive test is suggestive, but alone does not establish the strain as being hvKp. Likewise, a negative string test does not exclude the isolate as being hvKp.
- A number of word usage, typographical errors, and grammatical errors are present in the manuscript. A few examples include:
line 101- “paper”
line 191- “incomplete”
line 231- “mandator”
line 277- “Diagnostic” and “sustained”
Author Response
Thank you very much for your valuable comments. We have carefully revised our manuscript according to the observations and suggestions in the review. We have focused primarily on hvKp endogenous endophthalmitis, as you suggested, and expand the research to document other conditions that are associated with EKE, such as respiratory, urinary tract, osteomyelitis, cellulitis post spider bite
Major issues:
in the introduction we have discussed the differences between cKp and hvKp
we add also a paragraph about plasmid related multidrug resistance
we have removed the section focusing in KPLA
We have largely expanded the section regarding EKE, especially the management of EKE
Minor issues.
- Lines 183-189. Please don’t confuse colonization with entry for the reader. Colonization appears to be the first step requisite for infection, but colonization doesn’t necessarily lead to infection. Oropharyngeal colonization could lead to infection with aspiration, but it is still unclear what the mechanism of entry is via the GI tract since many individuals subsequently infected have no apparent GI pathology. Entry through micro-breaks in the skin, similar to Staphylococcus aureus is also a theoretical possibility.
Response: Thank you for the observation. We have corrected accordingly.
- lines 199-200. The sensitivity and specificity of the string test is not optimal. A positive test is suggestive, but alone does not establish the strain as being hvKp. Likewise, a negative string test does not exclude the isolate as being hvKp.
Response: I agree and we rephase the sentence. The message we wanted to send is that “the string test is an unexpensive tool, easy to perform in every practice, which can strongly suggest the involvement of a hypervirulent strain of Klebsiella pneumoniae. However, clinicians should be aware that a negative string test does not exclude the presence of hypervirulent Klebsiella pneumoniae in cases with invasive syndrome.
WE have corrected the spelling errors
Thank you again for your time and kind observations. We do hope that in this revised form you will find it ready for publication
Reviewer 2 Report
This is a review article on the importance of KP in the pathogenesis of hepatic abscess and endophthalmitis. KP is well-known on its virulence in the hepatic liver abscess in diabetic patients due to its thick capsule. The review tried to organized its pathogenic factors, treatment, as well as the incidence of liver abscess and endophthalmitis. However, the structure of the manuscript is very poor which make the reader hard to follow. Moreover, there are many spelling and grammatically errors in the entire manuscript. Comments Abstract Line 26: …..varied between 3.4% and 12.6%, with a media of 6.5%. What does the word “media” mean in this sentence? I suppose it is median rather then media. Line 28: a blank between agentwas. Line 29: “wild type antibiotic resistance. What is “wild type” antibiotic resistance? Introduction Line 43: ….located at the biliary ducts 50-70%. It does not make sense to have 50-70% at the end of the sentence. Results Figure 1 is not required. It does not provide any useful information in this specific topic. Section 3.1.1 It would be better to separate the epidemiology and risk factor. Line 145, Another section 3.1? There are two 3.1 sections. Line 152, Complement to complement Line 158 – 161. The author should also include the detection method for K2. Line 164. What is “tissular protein”? Line 168. It is quite strange to include a antibiotic susceptibility subsection below the 3.1. Moreover, what is “wild type” resistance? the author should define the “wild-type” resistance. Line 175, what is ESBL? And the author should describe in detail on how wang et al found that ESBL is an independent factor recurrence of KPLA. It is quite confusing to only place a sentence here and explain nothing. Line 179 I found this section did not provide enough information for me to realize what is the “specific features of KPLA”. I suggest the author remove this section or include those information to other sections. Line 201 what is polymerase chain reactions sequencing? Line 205 KP strains….. this sentence is not related to specific features of KPLA. Line 216 earl? Early Line 267. Anti-INF3? As for the section of EKE (section 3.5) I suggested the author organized following this order Incidence diagnostic Symptoms Risk factors Treatment Line 286. Broken sentence Line 288. …. vision more than 6/60 in 13.6% of cases. It is not a sentence that is easy to read. Line 301 The discussion section is not a discussion The entire section is just another section that summarized the findings of the references. The entire section should be completely re-written. Line 303. Could described. Could describe. There are many more grammatically errors and misleading sentences in the entire manuscript. Finally, the citation should be careful as well. It is quite strange to cite 15 different papers to proof a single point. For example, you cite references 6-20 on lines 255, 267. The citations should be specific.Author Response
Dear reviewer, thank you very much for your valuable comments. We have performed extensive English editing, to correct the spelling and typing errors. We also reorganized our manuscript according to your recommendations and the other reviewer's recommendations. We hope in this revised version you will find it ready to be published
Comments
Abstract Line 26: …..varied between 3.4% and 12.6%, with a media of 6.5%. What does the word “media” mean in this sentence? I suppose it is median rather then media.
Response: We have corrected with “median”
Line 28: a blank between agentwas.
Response: We have corrected the typing error.
Line 29: “wild type antibiotic resistance. What is “wild type” antibiotic resistance?
Response: “wild type” is the natural antibiotic resistance of bacteria, that is encountered in community acquired infection, as opposed to the modified, extended resistance characteristic for hospital acquired infections. We added this explanation, both in abstract section and in the main text
Introduction Line 43: ….located at the biliary ducts 50-70%. It does not make sense to have 50-70% at the end of the sentence.
Response: we have corrected the sentence
Results Figure 1 is not required. It does not provide any useful information in this specific topic.
Response: Based on my previous knowledge, PRISMA flowchart is required by many journals for systematic reviews. If Editor agree, I will remove it or submit as supplemental materials
Section 3.1.1 It would be better to separate the epidemiology and risk factor.
We performed the required changes
Line 145, Another section 3.1? There are two 3.1 sections.
We have corrected the numbering mistake.
Line 152, Complement to complement
Line 158 – 161. The author should also include the detection method for K2.
Line 164. What is “tissular protein”?
We replace it with “Fe-binding proteins”
Line 168. It is quite strange to include a antibiotic susceptibility subsection below the 3.1. Moreover, what is “wild type” resistance? the author should define the “wild-type” resistance.
We defined “wild type resistance”, as natural antibiotic resistance, usually encountered in community acquired infections. We also remove the subsection title, as all the described characteristics are part of the microbiological characteristics of hvKp
Line 175, what is ESBL? And the author should describe in detail on how wang et al found that ESBL is an independent factor recurrence of KPLA. It is quite confusing to only place a sentence here and explain nothing.
We defined ESBL as extended spectrum beta lactamases. We also developed the paragraph as required and move the entire section in the Treatment subsection
Line 179 I found this section did not provide enough information for me to realize what is the “specific features of KPLA”. I suggest the author remove this section or include those information to other sections.
We have removed the section as required. Some information was maintained in other sections.
Line 201 what is polymerase chain reactions sequencing?
Line 205 KP strains….. this sentence is not related to specific features of KPLA.
We removed this sentence
Line 216 earl? Early
We have corrected the spelling error
Line 267. Anti-INF3?
We added an explanatory sentence. IFN-3 signify type 3 interferon, which is considered to be involved in neutrophil response, hepatic macrophage activity and mucosal immunity. The identification of the anti-IFN3 in the case reported by Castle et al may explain the poor outcome, with bilateral vision loss, in a previously healthy patient
As for the section of EKE (section 3.5) I suggested the author organized following this order Incidence diagnostic Symptoms Risk factors Treatment
We reorganized the section as required and enlarged this section, especially the management of EKE
Line 286. Broken sentence
We have corrected.
Line 288. …. vision more than 6/60 in 13.6% of cases. It is not a sentence that is easy to read.
Line 301 The discussion section is not a discussion The entire section is just another section that summarized the findings of the references. The entire section should be completely re-written.
We have re-written the entire discussion section. We hope in this new form you will find it suitable for publication
Line 303. Could described. Could describe.
We have corrected
Finally, the citation should be careful as well. It is quite strange to cite 15 different papers to proof a single point. For example, you cite references 6-20 on lines 255, 267. The citations should be specific.
We change the references in the 2 locations to be more specific for the findings in associated pathology
Round 2
Reviewer 2 Report
I have no further comments.